# Pre-exposure prophylaxis implementation during incarceration: Perspectives of formerly incarcerated men and women

**Elizabeth Anna Banyas**[1], **Madelaine F. Castleman**[1], **Husnah A. Rahim**[1], **Eunice Okumu**[2], **Becky L. White**[3]*

**1** University of North Carolina at Chapel Hill, Chapel Hill, North Carolina, United States of America, **2** UNC Center for AIDS Research, University of North Carolina, Chapel Hill, NC, United States of America, **3** Department of Medicine, University of North Carolina at Chapel Hill, Chapel Hill, North Carolina, United States of America

* bls@med.unc.edu

## Abstract

The HIV prevalence is higher among individuals involved in the United States (U.S.) correctional system than those in general population. Despite this, people in prisons or other closed settings have poor access to the most effective biomedical prevention tool, HIV pre-exposure prophylaxis (PrEP). The purpose of this study was to explore the attitudes and beliefs of PrEP initiation in correctional facilities amongst individuals formerly in prisons or other closed settings. We conducted 13 in-depth qualitative interviews with recently released (from incarceration) clients participating in a formerly incarcerated transition program in the southern United States. We identified several themes from our interviews. These themes were 1) PrEP information during incarceration; 2) Risky behaviors during incarceration and post-release; 3) Anticipated post-release challenges to accessing PrEP. Individuals formerly in prisons or other closed settings desire more education about PrEP during incarceration. They also want to learn more about HIV prevention measures. They believe that initiating PrEP in prison would allow a seamless transition to post-release PrEP programs. In conclusion, initiating PrEP during incarceration is one strategy to increase access to PrEP among the medically underserved criminal-justice population.

## Introduction

The HIV seroprevalence is about three times higher among people formerly in prisons or other closed settings than among the general U.S. population [1]. The prevalence of HIV in jails is estimated at 1.3% [2]. The HIV prevalence in federal and state prisons is approximately 1.1% [3]. Within the prison population, the HIV prevalence among both men and women ranges from .2% in Wyoming to 2.8% in Florida.

Most individuals formerly in prisons or other closed settings acquire HIV prior to incarceration [4]. Women who have experienced incarceration engage in high-risk transmission behaviors in the community such as unprotected transactional sex [5]. Men who have experienced incarceration reportedly engage in injection drug use and unprotected sex with multiple

**Editor:** Hamid Sharifi, HIV/STI Surveillance Research Center and WHO Collaborating Center for HIV Surveillance, Institute for Future Studies in Health, Kerman University of Medical Sciences, ISLAMIC REPUBLIC OF IRAN

**Data Availability Statement:** There are ethical restrictions on sharing a de-identified data set as the data contain potentially sensitive information. As such, we can not share these data publicly. The contact for the research ethics committee at UNC is ethicsandintegrity@unc.edu.

**Funding:** This study was funded by the University of North Carolina at Chapel Hill Center for AIDS Research (CFAR) under grant number P30AI05041 with grant recipient BLW. URL of funder website is https://unccfar.org/acknowledge-the-cfar/. The funders played no role in the study design, data collection and analysis, decision to publish, or preparation of the manuscript.

**Competing interests:** The authors have declared that no competing interests exist.

sexual partners [6]. These risk behaviors in both genders are compounded by substance use, poverty and homelessness resulting in a "cycle of unsafe behavior" [7–9].

However, HIV transmission during incarceration does occur [10, 11]. There are several behaviors that place individuals at risk for HIV during incarceration. One behavior is sexual intercourse between people in prisons or other closed settings, whether consensual or non-consensual [12]. In fact, sexual assault is reported to occur at rates as high as 40% among males in prisons or other closed settings in the United States [13]. Another risk behavior is the use of unsterilized needles for injection drug use or tattooing [14]. About 31% of people in prisons have injected drugs in American prisons [15], as prisons are a place where drug use is frequently initiated as a coping method [16]. However, HIV transmission during incarceration may be mitigated by the widespread use of antiretroviral therapies (ARTs) resulting in treatment as prevention [17].

Currently, the most effective biomedical method to prevent HIV transmission to date is the use of ARTs to prevent HIV transmission. This method, called HIV pre-exposure prophylaxis (PrEP), reduces HIV infection by greater than 90% among men and women who acquire HIV sexually and 70% among individuals who inject drugs [18]. However, it has not been widely adopted in U.S. prisons and jails [19]. Further, PrEP uptake has been low among people formerly in prisons or other closed settings—the communities where most HIV transmission occurs [18]. To address this critical need, we explored the early feasibility of linking individuals formerly in prisons or other closed settings to post-release PrEP care. During our qualitative exploration, themes emerged around PrEP use during incarceration. In this study, we further explored these PrEP themes and present our findings.

## Methods

### Design and setting

This study was part of a larger study exploring the feasibility of a formerly incarcerated program for pre-exposure prophylaxis services in North Carolina (NC) [20]. The NC Formerly Incarcerated Transition (FIT) Program is part of a national program called the Transitions Clinic Network [21]. The purpose of the Transitions Clinic Network is to link people formerly in prisons or other closed settings to primary care. The program is delivered by community health workers formerly in prisons or closed settings who have successfully re-entered into their local communities. In addition, the program partners with community health centers and re-entry organizations who provide primary care and social services respectively. People formerly in prisons or other closed settings who participate in the Transitions Clinic Network are more likely to be linked to primary care, less likely to use emergency services and less likely to be re-incarcerated [22].

The recruitment period of the parent study took place between March 1st, 2019 and December 31st 2019. During, clients formerly in prisons or other closed settings as well as community stakeholders (e.g. community health workers, community health center clinicians, re-entry personnel) who were part of the NC FIT program were interviewed. All participants provided informed written and verbal consent. Participants interviewed over the telephone were sent a written consent document via email prior to questioning. Interviewers read the approved IRB verbal consent form verbatim to participants at the start of the interview. Consent was NOT waived. In this study, we limit our analysis to the Formerly Incarcerated Transition program clients or FIT clients.

### Inclusion and exclusion criteria

As set by the parent study, FIT client participants had to be greater than 18 years of age and able to understand and speak English. Participants also had to be in the program post-release

(jail and/or prison) for at least 6 months. They had to be able to provide verbal and written consent. Participants were excluded who were unable to provide consent including those with severe mental illness requiring immediate treatment or with mental illness limiting their ability to participate. Recruitment and interviews occurred from March 2019 through July 2019. Participants were recruited through community health clinic flyers, community health workers and community health clinician personnel. All FIT client study participants were offered a $25 gift card as an incentive for participation.

This study was approved by the Office of Human Research Ethics Non-Biomedical Institutional Review Board at the University of North Carolina at Chapel Hill and the North Carolina Department of Adult Correction's Human subjects review committee, and the Department of Health and Human Services' Office for Human Research Protections (OHRP).

## Qualitative data analysis

A qualitative descriptive study was conducted. This pragmatic approach was chosen "because a straightforward description of a phenomenon [was] desired" [22]. In addition, we chose the qualitative descriptive design because our purpose was to inform intervention development.

We used Bowen's feasibility framework to develop our semi-structured interview guide. Bowen's feasibility framework asserts that there are eight general areas of potential feasibility (e.g., acceptability, adaptation) depending on the study objectives. In our interview guide, we included four of Bowen's domains (acceptability, demand, integration, and expansion) [23]. In addition, we added PrEP implementation domains (PrEP knowledge, PrEP barriers, and PrEP facilitators) to our interview guide.

All interviews were recorded and transcribed word for word by a professional transcription service. Initially, all of the authors read all of the transcripts. Then, we re-read the transcripts to identify potential themes and codes. We then reviewed these initial codes. From these codes, we produced key subthemes. We used thematic analysis to identify broad emergent themes (Table 1). Next, we identified quotes that were illustrative of the overarching themes. Finally, we reviewed the themes again before we defined and named them. The finale themes were categorized into PrEP information during incarceration, risky behaviors during incarceration, and PrEP initiation prior to release.

## Results

Our total sample consisted of 13 people formerly in prisons or other closed settings with an average age of 38.31 years (range 29–57) (Table 2).

A majority of participants were male (69%) and identified as Caucasian/white (62%). Most had been in jail over eleven times (62%) with a greater variation in the frequency of prison

**Table 1. Summary of thematic claims.**

| Education | Behaviors | Transition |
|---|---|---|
| • Inadequate understanding of PrEP, including purpose, use, and administration<br>• Willingness to learn more about PrEP<br>• More likely to adhere to PrEP if made aware upon incarceration<br>• Increasing PrEP education would reduce stigmas and biases against HIV | • A greater occurrence of risky HIV behaviors within incarcerated populations leads to higher seropositive rate<br>• - PrEP adherence may serve as a preventative measure for those not intending and those intending to engage in risky behaviors | • Developing a routine during incarceration would contribute to a seamless transition upon release<br>• Concerns of side effects would be assuaged<br>• Post-release stressors including finding housing, healthcare, and employment would become a priority over PrEP uptake |

**Table 2. Demographics of qualitative interview participants, North Carolina, 2019.**

| Age in years, mean (SD) | | 38.31 (10.2) |
|---|---|---|
| | | No. (%) |
| Gender | | |
| | Male | 9 (69) |
| | Female | 4 (31) |
| Race | | |
| | Caucasian/White | 8 (62) |
| | Black/African American | 3 (23) |
| | Other/Mixed | 2 (15) |
| Ethnicity | | |
| | Non-Hispanic | 11 (85) |
| | Hispanic | 2 (15) |
| Education | | |
| | Less than high school degree | 1 (8) |
| | High school degree | 6 (46) |
| | More than a high school degree | 6 (46) |
| Housing status | | |
| | House | 3 (23) |
| | Apartment | 2 (15) |
| | Halfway House | 2 (15) |
| | Homeless | 1 (8) |
| | Trailer | 2 (15) |
| | Other | 3 (23) |
| Transportation | | |
| | Car | 6 (46) |
| | Bus | 5 (38) |
| | Other | 2 (15) |
| Number of times in jail | | |
| | 1–5 | 4 (31) |
| | 6–10 | 1 (8) |
| | 11+ | 8 (62) |
| Number of times in prison | | |
| | 0 | 5 (38) |
| | 1 | 3 (23) |
| | 2+ | 5 (38) |
| Most recent incarceration release date | | |
| | Past 3 months | 7 (54) |
| | 3–6 months | 4 (31) |
| | 7 months + | 2 (15) |
| Heard of PrEP | | |
| | Yes | 4 (31) |
| | No | 9 (69) |
| Willing to take PrEP | | |
| | Yes | 5 (38) |
| | Maybe | 1 (8) |
| | If needed | 5 (38) |

(*Continued*)

**Table 2.** (Continued)

| | Does not know | 2 (15) |
|---|---|---|

Data of 13 clients of the North Carolina Formerly Incarcerated Transition program. Abbreviations include: SD, standard deviation; No., number; PrEP, pre-exposure prophylaxis

incarcerations. Five participants (38%) had never been in prison, but five participants had been to prison more than two times. Seven participants (54%) had been released from either prison or jail within the past three months.

With respect to awareness, nine participants indicated they did not know what PrEP is. Yet, five participants (38%) indicated that they are willing to take PrEP, and another five participants indicated that they would take PrEP if needed. None of the participants stated they were not willing to take PrEP, but instead reported "maybe" or that they "did not know."

We now present attitudes and beliefs as they relate to the implementation of PrEP in correctional facilities: These themes were 1) PrEP information during incarceration; 2) Risky behaviors during incarceration; and 3) PrEP initiation prior to release.

## PrEP information during incarceration

Clients formerly in prisons or other closed settings believed that they should be educated about PrEP and given PrEP related materials during incarceration. They also believed that exposure to PrEP during incarceration was essential. A 60-year-old male expressed, "I don't know about the next person, but I know first you need exposure to the idea. You need the information to get to you otherwise you're gonna continue to move off that side of your brain that always gives you routine. I would say, you know, there's no real barrier if it's readily available, but, you know, if you're not–you don't have no knowledge of it then you–that's the last thing on your mind."

Interviewed FIT clients also recommended regularly scheduled informational seminars on PrEP by community members. This was suggested from a 45-year-old male, "maybe having a representative go maybe to the prisons and jails and maybe speak with the caseworkers about setting up like a small seminar every so often to people that are within 60 days of getting out or something like that. . .a seminar would be like perfect." Another 34-year-old male participant shared that people in prisons or other closed settings would be less likely to adhere to PrEP upon release if they were not made aware during incarceration. He stated, "a lot of people are not just going to get out and be like, let me go learn about this pill, just because I'm out of jail."

## Risky behaviors during incarceration

FIT client participants stated that high risk behaviors occur during incarceration often in unsanitary conditions. A 36-year-old female participant explains, "you never know. They do dirty things in jail and [laughs] they could get it in there too." Additionally, the unsanitary behaviors and environment of correctional facilities is a concern for individuals. "I don't know. I mean, honestly, it probably would be a good thing to have when you're–to take when you're in jail just because of all–it's very–it's not clean and if you have a cut or how many people are raped in jail all the time, you just never know when it could be transferred to you.

During incarceration, some individuals might engage in high-risk HIV transmission activities more than they would in the community. A 46-year-old female participant says, "you might find yourself in a certain situation where you wouldn't normally find yourself in and might make decisions that you wouldn't normally make and, you know, so you might—the

medication might come in handy 'cause—you know? And also, you know, a lot of times there's—take place and bad things happen to good people or whatever. You know what I mean"?

## PrEP initiation prior to release

FIT clients believed that initiating PrEP during incarceration would allow people in prisons or other closed settings time to routinize PrEP medication taking behavior. A 57-year-old male participant explained that "...if you've already started the regimen...it'll make the transition from there to out here a lot more easier to continue... 'Cause once you get out you gonna hit the floor running...things that you may have planned to do you don't execute. So, if you start them on a regimen prior it could be like second nature to do." Further, a 40-year-old male elucidated, "I think that transitioning and stuff like that should be done before people come out of prison, and the sad thing is a lot of people don't understand that–a lot of organizations, the government, society doesn't understand that people need–especially after doing a long time, they need to have that transition before." While in correctional facilities, participants explain they would have already initiated PrEP. They would learn to implement a novel medication regimen and adjust to the new routine. A 46-year-old female FIT client participant details with a simile, "...you could already have [PrEP] instilled into your daily function... it would be like brushing your teeth. It's just something you always do every single day...or eating breakfast. Drinking coffee. You know, something that's just already there." The same participant furthers with, "I think [PrEP implementation in correctional facilities] probably would be a good idea just to get in the practice of taking it and having it because that way, it'll already be sealed...it'll already be a function and a daily routine." However, some participants had concerns of potential side effects of the new PrEP medication. They suggest starting PrEP while in a controlled and monitored environment would assuage fears and promote adherence upon release. FIT client, a 32-year-old female advised, "I would [implement PrEP] before the release because then, honestly, when [incarcerated individuals] get out, they're probably gonna be taking other kinds of drugs so if they do have a side effect, you don't know if it's from that or if it's from a different drug. And so, I think it would benefit to do it while they're in jail so that if there is any type of side effect, you know that that's what it's from." A 36-year-old male participant also suggested, "...going in and getting it in their system so that their body doesn't have any kind of reactions to it. Get them comfortable taking something every day so that they won't forget to take it when they get out."

## Discussion

People in prisons or other closed settings expressed a need for educational tools to formulate a concrete understanding of the benefits of implementing PrEP upon incarceration. Educational awareness, including discussions from licensed professionals concerning HIV risk behaviors as well as the benefit of initial PrEP as an HIV preventative measure was discussed as a priority. Participants also expressed concern regarding education pertaining to the risk of HIV in the presence of currently incarcerated populations, and what risky behaviors increase HIV transmission. FIT clients are aware of the prevalence of high-risk behaviors present within corrective facilities. And subsequently, they express a desire to protect themselves during this period by taking PrEP. Additionally, educational parameters concerning general knowledge about the use of PrEP and its benefits, especially for currently incarcerated individuals, were expressed. Generally, there was a mutual agreement that educational will increase the likelihood of PrEP use.

Correctional facilities differ from environments in the outside world, forcing individuals to adjust within a short amount of time. People in prisons or other closed settings express how risky behaviors occur during incarceration. They state that the environment of the correctional facilities leaves them in a helpless state, causing them to participate in some activities that they would not normally participate in. Due to these conditions, they see PrEP as a method to protect themselves in a correctional environment.

Participants have a positive attitude towards implementing PrEP in correctional facilities. Therefore, participants explain that ensuring post-release PrEP adherence would require PrEP initiation during incarceration. Correctional medical staff could address participant concerns about PrEP side effects. Start-up symptoms would be cared for, regardless of if incarcerated individuals have medical insurance. Also, participants report they would more easily adhere to a medication regimen while in correctional facilities because dosing schedules, counseling, and support systems would be routinized. Participants explain that these PrEP developed habits would be more likely to carry over post-release.

Once released, these individuals are faced with a multitude of issues that may deter them from putting their health at a priority. Post-release, participants cited concerns surrounding access to healthcare and new stressors of daily life including familial obligations, stigma, and lack of resources. Further, many explain that implementing PrEP post-release would not be a priority. Rather, obtaining a job, finding shelter, and taking care of family would be a priority. Further, incarceration leads to post-release "disenfranchisement." This disenfranchisement worsens assess to housing and employment and other social determinants of health. Subsequently, their risk of HIV increases after release [20]. Therefore, the implementation of PrEP within correctional facilities would need to be coupled with post-release PrEP services addressing the social determinants of health.

## Limitations and strengths

There are several limitations to this study. First, our study sample consisted of clients formerly in prisons or other closed settings in a post-release transition program within a single state. Therefore, the themes elicited may not be salient for individuals not part of a post-release transition program. However, we believe that our findings will resonate with individuals formerly in prisons or other closed settings throughout the U.S. Second, our study was not designed to explore the potentially diverse perspectives of PrEP use during incarceration among key populations at high HIV risk (e.g. black men who have sex with men, transgender women, persons who inject drugs). Future research should consider additional perspectives.

A strength of our study is that it is based in the geographical region (southern) which has some of the lowest levels of PrEP users compared to the need in the U.S. [21]. Further, our qualitative exploration emerged from themes salient for the FIT clients, not predetermined topics important for researchers.

## Conclusion

Increasing PrEP uptake among individuals involved in the criminal justice system is crucial to ending the HIV epidemic. In our study, individuals in prisons or other closed settings are receptive to the implementation of PrEP in correctional facilities. FIT clients suggest starting PrEP while incarcerated. They believe PrEP initiation during incarceration may increase post-release PrEP adherence. In conclusion, HIV prevention for criminal justice involved individuals is crucial to ending the HIV epidemic in the U.S. This paper is a call to action to expand access to PrEP from the perspectives of people in the U.S. criminal justice setting to end the U.S. HIV epidemic.

## Acknowledgments

The authors would like to thank the North Carolina Formerly Incarcerated Transitions (FIT) clients who agreed to participate in the research, without whom this study would not be possible.

## Author Contributions

**Conceptualization:** Elizabeth Anna Banyas, Madelaine F. Castleman, Husnah A. Rahim.

**Data curation:** Becky L. White.

**Formal analysis:** Elizabeth Anna Banyas, Madelaine F. Castleman, Husnah A. Rahim.

**Funding acquisition:** Becky L. White.

**Methodology:** Elizabeth Anna Banyas, Madelaine F. Castleman, Husnah A. Rahim, Eunice Okumu.

**Project administration:** Elizabeth Anna Banyas.

**Supervision:** Elizabeth Anna Banyas, Becky L. White.

**Visualization:** Elizabeth Anna Banyas, Becky L. White.

**Writing – original draft:** Elizabeth Anna Banyas, Madelaine F. Castleman, Husnah A. Rahim.

**Writing – review & editing:** Elizabeth Anna Banyas, Eunice Okumu, Becky L. White.

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
