## [Decision Letter · Decision Letter 0]

9 Jun 2024

PONE-D-24-11121Pre-exposure prophylaxis implementation during incarceration: perspectives of formerly incarcerated men and womenPLOS ONE

Dear Dr. White,

Thank you for submitting your manuscript to PLOS ONE. After careful consideration, we feel that it has merit but does not fully meet PLOS ONE’s publication criteria as it currently stands. Therefore, we invite you to submit a revised version of the manuscript that addresses the points raised during the review process.

We look forward to receiving your revised manuscript.

Kind regards,

Hamid Sharifi, Professor

Academic Editor

PLOS ONE

Journal Requirements:

Additional Editor Comments:

Dear Authors,

Thanks so much for submitting this work to PLOS ONE. Both reviewers have recommended to accept manuscript. Howereve, before final decision, I have a few comments that should be apply.

1- The structure of the paragraphs should be improved. For example, the first two paragraphs of the introduction are only one sentece. Or the second last paragraph of the introduction should be impowered.

2- There is no need to add sutitles into the discussion.

3- Please check and fix the references based on PLOS ONE format.

Reviewers' comments:

Reviewer's Responses to Questions

**Comments to the Author**

1. Is the manuscript technically sound, and do the data support the conclusions?

Reviewer #1: Partly

Reviewer #2: Yes

2. Has the statistical analysis been performed appropriately and rigorously? 

Reviewer #1: Yes

Reviewer #2: Yes

3. Have the authors made all data underlying the findings in their manuscript fully available?

Reviewer #1: Yes

Reviewer #2: Yes

4. Is the manuscript presented in an intelligible fashion and written in standard English?

Reviewer #1: Yes

Reviewer #2: Yes

5. Review Comments to the Author

Reviewer #1: Thank you for your time and work in this very important area, and allowing me the opportunity to review your work.

I would advise that the author(s) add the transcript(s) of conversations as appendices to allow further context to the conversation. Despite the quotes being impactful and articulated well throughout, I feel that some of the power or context of the words of individuals may have been lost or missed in the bulk of the text.

This is an incredibly insightful written piece and it is very interesting to hear the views of those who are, or could be, directly affected - I appreciate that some of the conversations may have been triggering for individuals. Given the cultural differences outside of the U.S., it would be a great opportunity to build and expand on this piece of work in other countries and gain further insight into the understanding and provision of PrEP.

Reviewer #2: 1. Is the manuscript technically sound, and do the data support the conclusions?

The manuscript appears technically sound. The study is based on qualitative research and includes detailed descriptions of the methodology, data collection, and analysis processes. The conclusions drawn by the authors are supported by the data, which are well-documented and aligned with the study's objectives and findings. The discussion section logically connects the results to the broader context of HIV prevention among formerly incarcerated individuals, suggesting that the data adequately support the conclusions made.

2. Has the statistical analysis been performed appropriately and rigorously?

The manuscript employs qualitative research methods, and while it does not involve traditional statistical analysis, the qualitative data analysis is performed rigorously. Themes were derived from participant responses, and the authors followed a systematic approach to coding and interpreting the data. This method is appropriate given the research question and study design, ensuring the analysis is thorough and valid within the context of qualitative research.

3. Have the authors made all data underlying the findings in their manuscript fully available?

The manuscript includes a Data Availability Statement indicating that there are ethical restrictions on sharing a de-identified data set because the data contain potentially sensitive information. However, the contact information for the research ethics committee at UNC is provided for researchers who meet the criteria for access to confidential data. This meets the journal's requirements for data availability under conditions where full public sharing is not possible.

4. Is the manuscript presented in an intelligible fashion and written in standard English?

The manuscript is presented clearly and is written in standard English. The structure is logical, with a well-defined introduction, methodology, results, discussion, and conclusion. The language is professional and accessible, making the manuscript intelligible to a broad academic audience. The authors effectively communicate their findings and their significance to the field.

Overall, the manuscript is well-prepared and adheres to the standards expected for publication.

6. PLOS authors have the option to publish the peer review history of their article (what does this mean?). If published, this will include your full peer review and any attached files.

Reviewer #1: **Yes: **Geraint Jones

Reviewer #2: **Yes: **Ayalew Aklilu Haile

---

## [Author Response · Author response to Decision Letter 0]

11 Jul 2024

Dear Editor, Reviewers,

We thank you for your fair reviews and constructive suggestions. The authors appreciate the opportunity to submit a minor revision and hope that the manuscript is now suitable for publication. We attached a file titled "Response to Reviewers" where we respond to each comment in bold. Line numbers within refer to changes made in the document titled “Manuscript with Track Changes.”

Sincerely, 

On behalf of all authors, 

Elizabeth Banyas

---

## [Editor Report · Decision Letter 1]

15 Jul 2024

PONE-D-24-11121R1Pre-exposure prophylaxis (PrEP) implementation during incarceration: perspectives of formerly incarcerated men and womenPLOS ONE

Dear Dr. White,

Thank you for submitting your manuscript to PLOS ONE. After careful consideration, we feel that it has merit but does not fully meet PLOS ONE’s publication criteria as it currently stands. Therefore, we invite you to submit a revised version of the manuscript that addresses the points raised during the review process.

We look forward to receiving your revised manuscript.

Kind regards,

Hamid Sharifi, Professor

Academic Editor

PLOS ONE

Journal Requirements:

Additional Editor Comments:

Dear Authors,

Thanks for resubmit the revised work. Before the final decision, please consider and apply these two comments:

1- Using this new released guideline, please revise some words like prisoners or incarcerated people.

2- You need also share a file including the comments and also your responses.

Best Regards

Hamid Sharifi

Professor in Epidemiology

---

## [Author Response · Author response to Decision Letter 1]

15 Sep 2024

Dear Editors and Reviewers,

We thank you for your fair reviews and constructive suggestions. The authors appreciate the opportunity to submit a minor revision and hope that the manuscript is now suitable for publication. Below, we respond to each comment in bold. Line numbers refer to line numbers within the document titled “Manuscript with Track Changes.”

Editor Comment #1 (from first revision): The structure of the paragraphs should be improved. For example, the first two paragraphs of the introduction are only one sentence. Or the second last paragraph of the introduction should be impowered.

The structure of the first paragraph within the introduction was adjusted [39]. We connected the first, single, standalone sentence within the following paragraph.

We also divided the second paragraph of the introduction into two paragraphs. The new second paragraph focuses on HIV acquisition and risk prior to incarceration. The new third paragraph is focused on HIV transmission in correctional settings as previous. 

The second to last paragraph within the introduction was improved in three ways. The revised paragraph introduces the acronym ‘‘PrEP”.) The reference to Zambian prisons was removed as it does not relate to the scope of this paper . We explicitly state there is a gap between efficacy and availability of PrEP. Citation numbers were adjusted accordingly within the manuscript and reference list.

Editor Comment #2 (from first revision): There is no need to add subtitles into the discussion.

All subtitles within the discussion are removed. 

Editor Comment #3 (from first revision): Please check and fix the references based on PLOS ONE format.

All references now follow the PloS ONE format.

Editor Comment #1 (from second revision): Using this new released guideline, please revise some words like prisoners or incarcerated people.

We appreciate you for providing the UNAIDS terminology guidelines. We went through the manuscript and updated any terms no longer in use to preferred terms.

Editor Comment #2 (from second revision): The previous submitted file did not include the previous comments and also the responses of the authors. I need a cover letter including the previous comments and also the responses of the author how they did apply the comments.

We hope that this document containing comments and responses meets the standards of PLoS One.

Reviewer Comment #1: Thank you for your time and work in this very important area, and allowing me the opportunity to review your work. I would advise that the author(s) add the transcript(s) of conversations as appendices to allow further context to the conversation. Despite the quotes being impactful and articulated well throughout, I feel that some of the power or context of the words of individuals may have been lost or missed in the bulk of the text.

Thank you for your comment. To meet PloS ONE’s requirements for data availability, our submission includes a Data Availability Statement. The statement explains there are ethical restrictions on publishing a de-identified data set to the public. Thus, the authors believe the transcripts should not be included within an appendix to protect potentially sensitive information. However, researchers who meet the criteria to access confidential data are encouraged to contact the research ethics committee at UNC Chapel Hill to obtain access.

Sincerely yours,

Elizabeth Banyas, Madelaine Castleman, Husnah Rahim, and Becky White

---

## [Editor Report · Decision Letter 2]

17 Sep 2024

Pre-exposure prophylaxis (PrEP) implementation during incarceration: perspectives of formerly incarcerated men and women

PONE-D-24-11121R2

Dear Dr. White,

We’re pleased to inform you that your manuscript has been judged scientifically suitable for publication and will be formally accepted for publication once it meets all outstanding technical requirements.

Kind regards,

Hamid Sharifi, Professor

Academic Editor

PLOS ONE
---

## [Editor Report · Acceptance letter]

24 Sep 2024

PONE-D-24-11121R2 

PLOS ONE

Dear Dr. White, 

I'm pleased to inform you that your manuscript has been deemed suitable for publication in PLOS ONE. Congratulations! Your manuscript is now being handed over to our production team.

Kind regards, 

on behalf of

Dr. Hamid Sharifi 

Academic Editor

PLOS ONE